# A timeline of bacterial and archaeal diversification in the ocean

Carolina A Martinez-Gutierrez[1]*, Josef C Uyeda[1], Frank O Aylward[1,2]*

[1]Department of Biological Sciences, Virginia Tech, Blacksburg, United States; [2]Center for Emerging, Zoonotic, and Arthropod-borne Pathogens, Virginia Tech, Blacksburg, United States

**Abstract** Microbial plankton play a central role in marine biogeochemical cycles, but the timing in which abundant lineages diversified into ocean environments remains unclear. Here, we reconstructed the timeline in which major clades of bacteria and archaea colonized the ocean using a high-resolution benchmarked phylogenetic tree that allows for simultaneous and direct comparison of the ages of multiple divergent lineages. Our findings show that the diversification of the most prevalent marine clades spans throughout a period of 2.2 Ga, with most clades colonizing the ocean during the last 800 million years. The oldest clades – SAR202, SAR324, *Ca.* Marinimicrobia, and Marine Group II – diversified around the time of the Great Oxidation Event, during which oxygen concentration increased but remained at microaerophilic levels throughout the Mid-Proterozoic, consistent with the prevalence of some clades within these groups in oxygen minimum zones today. We found the diversification of the prevalent heterotrophic marine clades SAR11, SAR116, SAR92, SAR86, and Roseobacter as well as the Marine Group I to occur near to the Neoproterozoic Oxygenation Event (0.8–0.4 Ga). The diversification of these clades is concomitant with an overall increase of oxygen and nutrients in the ocean at this time, as well as the diversification of eukaryotic algae, consistent with the previous hypothesis that the diversification of heterotrophic bacteria is linked to the emergence of large eukaryotic phytoplankton. The youngest clades correspond to the widespread phototrophic clades *Prochlorococcus, Synechococcus,* and *Crocosphaera,* whose diversification happened after the Phanerozoic Oxidation Event (0.45–0.4 Ga), in which oxygen concentrations had already reached their modern levels in the atmosphere and the ocean. Our work clarifies the timing at which abundant lineages of bacteria and archaea colonized the ocean, thereby providing key insights into the evolutionary history of lineages that comprise the majority of prokaryotic biomass in the modern ocean.

## eLife assessment

This **important** paper addresses the challenging problem of dating the origin of several groups of marine microorganisms. The analyses are **solid**, with various test of clock models and time calibrations used; however, given the uncertainty of many of the dates used to anchor ancient geological events, further studies are needed to support or refute the hypotheses put forth in this paper. Despite some methodological concerns, this work is a commendable attempt at an extremely difficult problem and will be of broad interest to microbiologists, geologists, and evolutionary biologists.

## Introduction

The ocean plays a central role in the fluxes and stability of Earth's biogeochemistry (*Dontsova et al., 2020*; *Field et al., 1998*; *Falkowski et al., 1998*). Due to their abundance, diversity, and physiological versatility, microbes mediate the vast majority of organic matter transformations that underpin

*For correspondence:
cmartinez@vt.edu (CAM-G);
faylward@vt.edu (FOA)

Competing interest: The authors declare that no competing interests exist.

higher trophic levels (*Brown et al., 2014*; *Mason et al., 2009*). For example, marine microorganisms regulate a large fraction of the organic carbon pool (*Ducklow and Doney, 2013*), drive elemental cycling of nutrients like nitrogen (*Zehr and Kudela, 2011*), and participate in the ocean-atmosphere exchange of climatically important gasses (*Vila-Costa et al., 2006*). Starting in the 1980s, analysis of small-subunit ribosomal RNA genes began to reveal the identity of dominant clades of bacteria and archaea that were notable for their ubiquity and high abundance, and subsequent analyses high-lighted their diverse physiological activities in the ocean (*Giovannoni and Stingl, 2005*). Phylogenetic studies showed that these clades are broadly distributed across the Tree of Life (ToL) and encompass a wide range of phylogenetic breadths (*Giovannoni and Stingl, 2005*). Cultivation-based studies and the large-scale generation of genomes from metagenomes have continued to make major progress in examining the genomic diversity and metabolism of these major marine clades, but we still lack a comprehensive understanding of the evolutionary events leading to their origin and diversification in the ocean.

Several independent studies have used molecular phylogenetic methods to date the diversification of some marine microbial lineages, such as the ammonia-oxidizing archaea of the order *Nitrososphaerales* (Marine Group I [MGI]) (*Ren et al., 2019*; *Yang et al., 2021*; *Zhang et al., 2021*), picocyanobacteria of the genera *Synechococcus* and *Prochlorococcus* (*Sánchez-Baracaldo, 2015*; *Sánchez-Baracaldo et al., 2019*; *Zhang et al., 2021*), and marine alphaproteobacterial groups that included the SAR11 and Roseobacter clades (*Luo et al., 2013*). Differences in the methodological frameworks used in these studies often hinder comparisons between lineages, however, and results for individual clades often conflict (*Ren et al., 2019*; *Sánchez-Baracaldo, 2015*; *Yang et al., 2021*; *Zhang et al., 2021*). Moreover, it has been difficult to directly compare bacterial and archaeal clades due to the vast evolutionary distances between these domains. For these reasons, it has remained challenging to evaluate the ages of different marine lineages and develop a comprehensive under-standing of the timing of microbial diversification events in the ocean and their relationship with major geological events throughout Earth's history.

To clarify the timing at which major lineages of bacteria and archaea diversified into the ocean, we developed an approach that leverages a multi-domain phylogenetic tree that allows for simultaneous dating of all major marine lineages. This method allows us to directly compare the ages of divergent lineages across the ToL and subsequently reconstruct a timeline in which these groups evolved into the ocean. Moreover, we can also map the acquisition of different protein families onto this phylogeny and thereby infer the genes that were gained by these marine lineages at the time of their emergence. Altogether, our study provides a comprehensive framework that sheds light on watershed events in the history of life on Earth that have given rise to contemporary biodiversity and biogeochemical dynamics in the ocean.

## Results and discussion

To begin analyzing the diversification of marine lineages of bacteria and archaea, we constructed a multi-domain phylogenetic tree that allowed us to directly compare the origin of 13 planktonic marine bacterial and archaeal clades that are notable for their abundance and major roles in marine biogeo-chemical cycles (*Figure 1*). We based tree reconstruction on a benchmarked set of marker genes that we have previously shown to be congruent for inter-domain phylogenetic reconstruction (*Martinez-Gutierrez and Aylward, 2021*; details in 'Materials and methods,' *Supplementary file 2*). Our phylo-genetic framework included non-marine clades for phylogenetic context, and overall it recapitulates known relationships across the ToL, such as the clear demarcation of the Gracilicutes and Terrabacteria bacterial superphyla and the basal placement of the *Thermatogales* within Bacteria (*Coleman et al., 2021*; *Martinez-Gutierrez and Aylward, 2021*; *Figure 1*). To gain insight into the geological land-scape in which these major marine clades first diversified, we performed a Bayesian relaxed molecular dating analysis on our benchmarked ToL using several calibrated nodes (*Figure 1* and *Table 1*).

Due to the limited representation of microorganisms in the fossil record and the difficulties to asso-ciate fossils to extant relatives, we employed geochemical evidence as temporal calibrations (*Figure 1* and *Table 1*). Moreover, because of the uncertainty in the length of the branch linking bacteria and archaea, the crown node for each domain was calibrated independently. We used both the age of the presence of liquid water (as approximated through the dating of zircons; *Valley et al., 2014*) as well as the most ancient record of biogenic methane (broadly used as evidence of life on Earth; *Ueno et al.,*

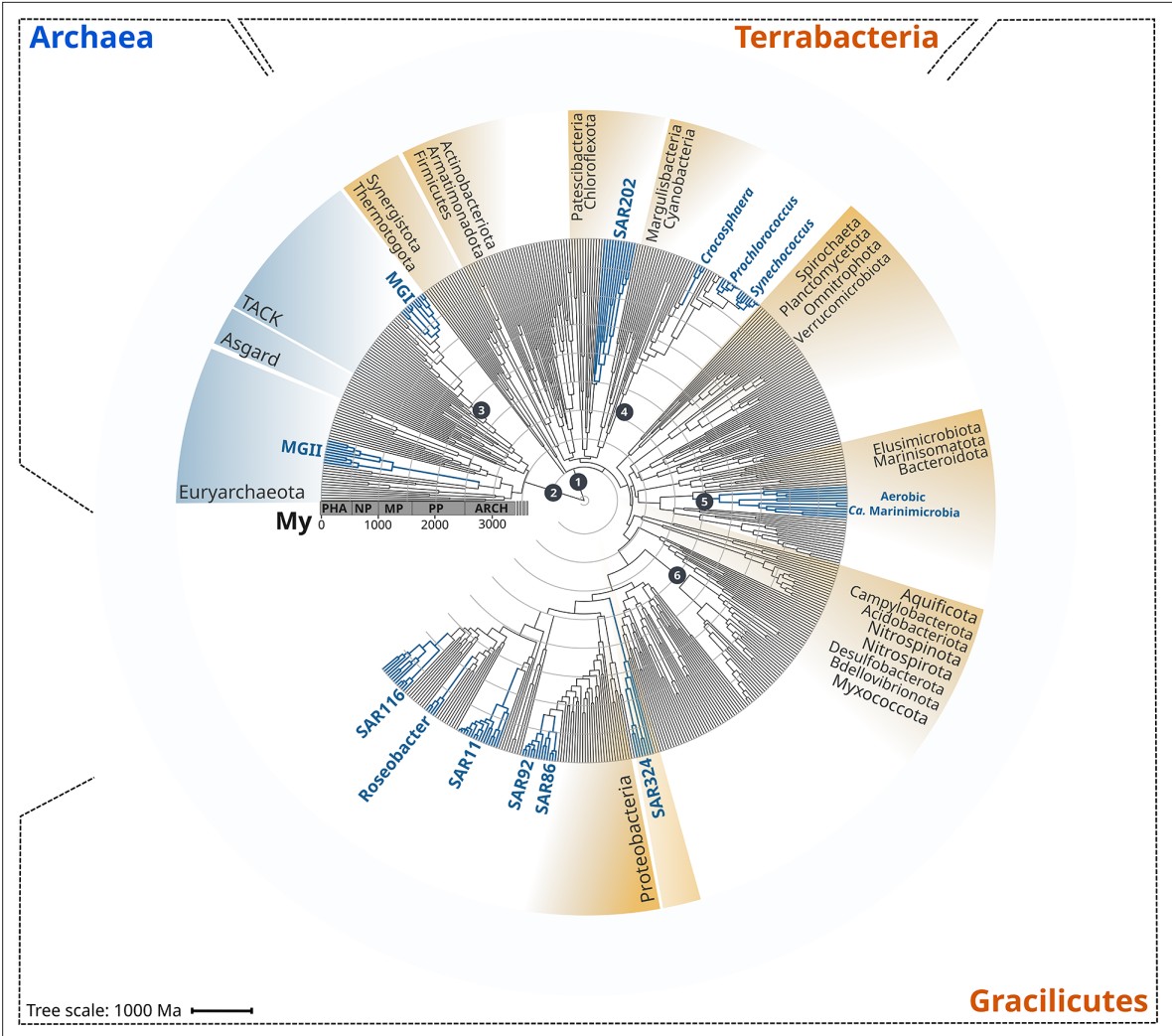

**Figure 1.** Rooted inter-domain Tree of Life used for molecular dating analyses. Maximum likelihood tree constructed with IQ-TREE v1.6.12 using the concatenation of 30 RNAP subunits and ribosomal protein sequences and the substitution model LG + R10. Blue labels represent the marine clades dated in our study. Dark gray dots show the temporal calibration used in our molecular dating analyses (*Table 1*). The marine clades shown are classified on the Genome Taxonomy Database (GTDB) as follows: MGII, *Poseidoniales;* MGI, *Nitrososphaerales;* SAR202, SAR202; *Crocosphaera, Crocosphaera; Prochlorococcus, Prochlorococcus; Synechococcus, Synechococcus;* Ca. Marinimicrobia, *Marinisomatia;* SAR324, SAR324; SAR86, *Oceanospirillales;* SAR92, *Porticoccaceae;* SAR11, *Pelagibacterales;* Roseobacter, *Rhodobacteraceae;* SAR116, *Puniceispirillaceae.* Abbreviations: PHA, Phanerozoic; NP, Neoproterozoic; MP, Mesoproterozoic; PP, Paleoproterozoic; ARCH, Archaean.

*2006*) as maximum and minimum prior ages for bacteria and archaea (4400 and 3460 My, respectively, *Figure 1* and *Table 1*). For internal calibration, we used the recent identification of non-oxygenic Cyanobacteria to constrain the diversification node of oxygenic Cyanobacteria with a minimum age of 2320 My, the estimated age for the Great Oxidation Event (GOE) (*Bekker et al., 2004*; *Holland, 2006*; *Holland, 2002*). Similarly, we calibrated the crown node of aerobic ammonia-oxidizing archaea, aerobic Ca. Marinimicrobia, and the nitrite-oxidizing bacteria with a maximum age of 2320 My (GOE estimated age) due to their strict aerobic metabolism. Despite geological evidence pointing to the presence of oxygen before the GOE, our Bayesian estimates indicate an overall consistency of the priors used (*Figure 2—figure supplement 2*), and we recovered the ancient origin of major bacterial and archaeal supergroups, such as Asgardarchaeota, Euryarchaeota, Firmicutes, Actinobacteria, and Aquificota (*Figure 2*). Moreover, the date we found for oxygenic Cyanobacteria (2611 My, 95% CI 2589–2632; *Figure 2*) is in agreement with their diversification happening before the GOE (*Ward et al., 2016*). Please see *Table 1* for a detailed explanation of all calibration dates used, together with our rationale for including each one.

**Table 1.** Temporal calibrations used as priors for the molecular dating of the main marine microbial clades.

See 'Materials and methods' for a detailed explanation of the calibrations used.

| Node | Calibration group | Minimum (My) | Maximum (My) | Evidence | Reference |
|------|------|------|------|------|------|
| 1,2 | Bacteria-Archaea Root | - | 4400 | Identification of the most ancient zircons showing evidence of liquid water. | *Valley et al., 2014* |
| 1,2 | Bacteria-Archaea Root | 3460 | - | Identification of the most ancient traces of methane. Minimum age for life on Earth. Calibration consistent with the most ancient fossils found to date (~3.5 Ga; *Walter et al., 1980*). | *Ueno et al., 2006* |
| 3 | Aerobic *Nitrososphaerales* | - | 2320 | Strict aerobic metabolism. | *Ueno et al., 2006* |
| 4 | Oxygenic Cyanobacteria | 2320 | - | Oxygenation of the atmosphere. The Great Oxidation Event has been associated with oxygenic Cyanobacteria. | *Bekker et al., 2004*; *Holland, 2006*; *Holland, 2002* |
| 5 | Aerobic *Ca.* Marinimicrobia | - | 2320 | Strict aerobic metabolism. | *Bekker et al., 2004*; *Holland, 2006*; *Holland, 2002* |
| 6 | Nitrite-oxidizing bacteria | - | 2320 | Strict aerobic metabolism. | *Bekker et al., 2004*; *Holland, 2006*; *Holland, 2002* |

Our Bayesian estimates suggest that the lineages that emerged earliest are SAR202, aerobic *Ca.* Marinimicrobia, SAR324, and the Marine Group II of the phylum Euryarchaeota (MGII). The most ancient clade was the SAR202 (2479 My, 95% CI 2465–2492 My), whose diversification took place near before the GOE (*Figure 2*). Given the broadly distributed aerobic capabilities of SAR202, the diversification of this clade before the GOE suggests that SAR202 emerged during an oxygen oasis proposed to have existed in pre-GOE Earth (*Anbar et al., 2007*; *Ossa Ossa et al., 2019*; *Reinhard and Planavsky, 2022*). The ancient pre-GOE origin of SAR202 is consistent with a recent study that proposed that this clade played a role in the shift of the redox state of the atmosphere during the GOE. SAR202 is able to partially metabolize organic matter through a flavin-dependent Baeyer–Villiger monooxygenase, thereby enhancing the burial of organic matter and contributing to the net accumulation of oxygen in the atmosphere (*Landry et al., 2017*; *Shang et al., 2022*). After the GOE, we detected the diversification of aerobic *Ca.* Marinimicrobia (2196 My, 95% CI 2173–2219 My), the SAR324 clade (1686 My, 95% CI 1658–1715 My), and the MGII clade (1184 My, 95% CI 1166–1202 My) (*Figure 2*). Although these ancient clades may have first diversified under the oxic conditions derived from the GOE, it has been suggested that the initial oxygenation of Earth was followed by a relatively rapid drop in ocean and atmosphere oxygen levels (*Alcott et al., 2019*; *Hodgskiss et al., 2019*; *Reinhard and Planavsky, 2022*). It is therefore likely that these clades diversified in the microaerophilic and variable oxygen conditions that prevailed during this period (*Bekker et al., 2004*; *Holland, 2006*; *Holland, 2002*). Indeed, the oxygen landscape in which these marine clades first diversified is consistent with their current physiology. For example, these groups are capable of using oxygen and other compounds as terminal electron acceptors (e.g., nitrate and sulfate), and several representatives are prevalent in modern marine oxygen minimum (OMZs) (*Pajares et al., 2020*; *Sheik et al., 2014*; *Thrash*

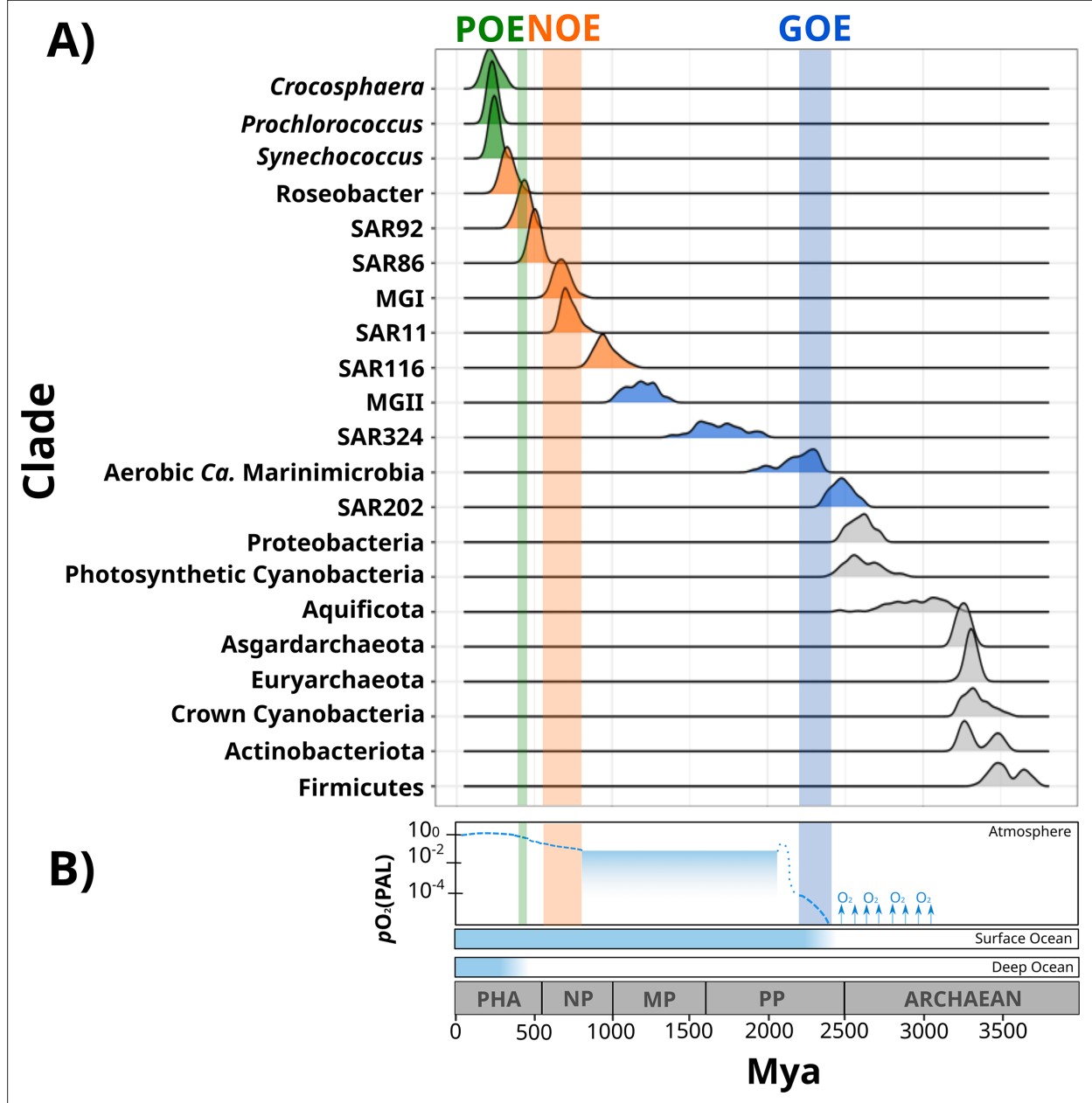

**Figure 2.** Dates of the diversification of marine microbial clades and the redox history of Earth's atmosphere, surface ocean, and deep ocean. (**A**) Ridges represent the distribution of 100 Bayesian dates estimated using a relaxed molecular clock and an autocorrelated model (see 'Materials and methods'). Ridges of marine clades were colored based on their diversification date: green, late-branching phototrophs; orange, late-branching clades; blue, early-branching clades. The timing of the diversification of major bacterial and archaeal superphyla is represented with gray ridges. Molecular dating estimates resulting from the uncorrelated model Uncorrelated Gamma Multiplies (UGAM) and the Autocorrelated Model CIR are shown in *Figure 2— figure supplement 3*. (**B**) Oxygenation events and redox changes across Earth's history. Panel adapted from previous work (Figure 1 of *Alcott et al., 2019*). Abbreviations: POE, Paleozoic Oxidation Event; NOE, Neoproterozoic Oxidation Event; GOE, Great Oxidation Event; Pha, Paleozoic; NP, Neoproterozoic; MP, Mesoproterozoic; PP, Paleoproterozoic.

The online version of this article includes the following figure supplement(s) for figure 2:

**Figure supplement 1.** Comparison of the age distribution of marine microbial clades using a Bayesian and a penalized likelihood approach for molecular dating.

**Figure supplement 2.** Estimated ages for calibrated nodes showing their suitability as priors for Bayesian molecular dating.

**Figure supplement 3.** Assessment of the role of molecular dating Bayesian model and calibrations on the diversification timing of marine microbial clades.

*et al., 2017*; *Ulloa et al., 2012*). The facultative aerobic or microaerophilic metabolism in these clades is potentially a vestige of the low oxygen environment of most of the Proterozoic Eon, and in this way OMZs can be considered to be modern-day refugia of these ancient ocean conditions. Of the clades that diversified as part of this early period, MGII and SAR324 show the youngest colonization dates, but we suspect that this may be due to the notably long branches that lead to the crown nodes of these lineages. These long branches are likely caused by the absence of basal-branching members of these clades – either due to extinction events or under-sampling of rare lineages in the available genome collection – that would have increased the age of these lineages if present in the tree.

According to our analysis, the next clades to diversify in the ocean are SAR116 (959 My, 95% CI 945–973 My), SAR11 (725 My, 95% CI 715–734 My), SAR86 (503 My, 95% CI 497–509 My), SAR92 (430 My, 95% CI 423–437 My), and Roseobacter (332 My, 95% CI 323–340 My) (*Figure 2*). The relatively late appearance of these heterotrophic lineages that are abundant in the open ocean today was potentially due to the low productivity and oxygen concentrations in both shallow and deep waters that prevailed in the Mid-Proterozoic (1800–800 My), a period previously described as the 'boring billion' (*Anbar and Knoll, 2002*; *Crockford et al., 2018*; *Hodgskiss et al., 2019*; *Planavsky et al., 2014*; *Tang et al., 2016*). The diversification of these clades may be indirectly associated with the Snowball event registered before the Neoproterozoic Oxidation Event (NOE, 800–540 My) (*Anbar and Knoll, 2002*; *Hoffman et al., 1998*; *Shields-Zhou and Och, 2011*), which was followed by an increased the availability of oxygen and inorganic nutrients in the ocean (*Anbar and Knoll, 2002*; *Butterfield, 2001*; *Porter, 2004*; *Shields-Zhou and Och, 2011*; *Vidal and Moczydłowska-Vidal, 1997*), and is also coincident with the widespread diversification of large eukaryotic algae during the Neoproterozoic (*Anbar and Knoll, 2002*; *Butterfield, 2001*; *Porter, 2004*; *Shields-Zhou and Och, 2011*; *Vidal and Moczydłowska-Vidal, 1997*). It is therefore plausible that an increase in nutrients as well as the broad diversification of eukaryotic plankton enhanced the mobility of organic and inorganic nutrients beyond the coastal areas, and increased the burial of organic matter that consequently led to the rise in atmospheric oxygen concentrations (*Knoll et al., 2006*; *Shields-Zhou and Och, 2011*). The scenario in which heterotrophic marine clades diversified in part as a consequence of the new niches built by marine eukaryotes has been previously proposed to have driven the diversification of the Roseobacter clade (*Luo et al., 2013*; *Luo and Moran, 2014*). The diversification timing of Roseobacter and other heterotrophic clades supports this phenomenon and suggests that the interaction with marine eukaryotes may have broadly influenced the diversification of prevalent lineages in the modern ocean. Similar to what we observed in MGII and SAR324, the Roseobacter clade shows a long branch leading to the crown node (*Figure 1*), suggesting that the diversification of this clade may have occurred earlier.

We also report the diversification of the chemolithoautotrophic archaeal lineage MGI into the ocean after the NOE (678 My, 95% CI 668–688 My) (*Figure 2*), which is comparable with the age reported by another independent study (*Yang et al., 2021*). This is consistent with an increase in the oxygen concentrations of the ocean during this period (*Reinhard and Planavsky, 2022*), a necessary requisite for ammonia oxidation. Moreover, the widespread sulfidic conditions that likely prevailed during the Mid-Proterozoic ocean may have limited the availability of redox-sensitive metals such as copper, which is necessary for ammonia monooxygenases (*Anbar and Knoll, 2002*; *Hatzenpichler, 2012*). It is therefore plausible that a low concentration of oxygen and limited inorganic nutrient availability before the NOE were limiting factors that delayed the colonization of AOA into the ocean.

The most recent lineages to emerge include the genera *Synechococcus* (243 My, 95% CI 238–247 My), *Prochlorococcus* (230 My, 95% CI 225–234 My), and the diazotroph *Crocosphaera* (228 My, 95% CI 218–237 My). Our results agree with an independent study that points to a relatively late emergence of the marine picocyanobacterial clades *Prochlorococcus* and *Synechococcus* (*Sánchez-Baracaldo, 2015*). Picocyanobacterial clades and *Crocosphaera* are critical components of phytoplanktonic communities in the modern open ocean due to their large contribution to carbon and nitrogen fixation, respectively (*Flombaum et al., 2013*; *Montoya et al., 2004*; *Scanlan et al., 2009*). For example, the nitrogen fixation activities of *Crocosphaera watsonii* in the open ocean today support the demands of nitrogen-starved microbial food webs found in oligotrophic waters (*Hewson et al., 2009*). The relatively late diversification of these lineages suggests that the oligotrophic open ocean is a relatively modern ecosystem. Moreover, the oligotrophic ocean today is characterized by the rapid turnover of nutrients that depends on the efficient mobilization of essential elements

through the ocean (*Karl, 2002*). Due to its distance from terrestrial nutrient inputs, productivity in the open ocean is therefore dependent on local nitrogen fixation, which was likely enhanced after the widespread oxygenation of the ocean that made molybdenum widely available due to its high solubility in oxic seawater (*Canfield et al., 2007*; *Scott et al., 2008*; *Wei et al., 2021*). Such widespread oxygenation was registered 430–390 My in an event referred to here as the Paleozoic Oxidation Event (POE; *Berner and Raiswell, 1983*; *Lenton et al., 2016*; *Sperling et al., 2015*; *Tostevin and Mills, 2020*; *Figure 2*). The increase of oxygen to present-day levels in the atmosphere and the ocean was potentially the result of an increment of the burial of organic carbon in sedimentary rocks due to the diversification of the earliest land plants (*Lenton et al., 2016*; *Planavsky et al., 2021*; *Reinhard and Planavsky, 2022*). The POE has also been associated with increased phosphorus weathering rates (*Bergman, 2004*; *Lenton et al., 2016*), global impacts on the global element cycles (*Dahl and Arens, 2020*), and an increase in the overall productivity of the ocean (*Planavsky et al., 2021*). The late diversification of oligotrophic-specialized clades after the POE therefore suggests that the establishment of the oligotrophic open ocean as we know it today would only have been plausible once modern oxygen concentrations and biogeochemical dynamics were reached (*Karl, 2002*; *Reinhard and Planavsky, 2022*).

To shed further light on the drivers that allowed their colonization into the ocean, we investigated whether the diversification of marine microbial clades was linked to the acquisition of novel metabolic capabilities. We broadly classified the different clades as early-branching clades (EBC), late-branching clades (LBC), and late-branching phototrophs (LBP) based on the general timing of their diversification (*Figure 2*). To identify the enrichment of gene functions at the crown node of each marine clade (*Figure 1*), we performed a stochastic mapping analysis on each of the 112,248 protein families identified in our genome dataset (*Supplementary file 1*). We compared our results with a null hypothesis distribution in which a constant rate model was implemented unconditionally of observed data (see 'Materials and methods'). Statistical comparisons of the stochastic and the null distribution show that each diversification phase was associated with the enrichment of specific functional categories that were consistent with the geochemical context of their diversification (*Figures 3 and 4*). For example, EBC gained a disproportionate number of genes involved in DNA repair, recombination, and glutathione metabolism, consistent with the hypothesis that the GOE led to a rise in reactive oxygen species that cause DNA damage (*Zaikowski et al., 2010*; *Khademian and Imlay, 2021*; *Masip et al., 2006*). Moreover, the EBC were enriched in proteins involved in ancient aerobic pathways, such as oxidative phosphorylation and the TCA cycle (*Figure 3*), as well as genes implicated in the degradation of fatty acids under aerobic conditions, such as the enzyme alkane 1-monooxygenase in MGII (*Supplementary file 4*). We also detected genes for the adaptation to marine environments, including genes for the anabolism of taurine (e.g., cysteine dioxygenase in MGII, *Supplementary file 4*), an osmoprotectant commonly found in marine bacteria (*McParland et al., 2021*). Our findings suggest that the diversification of EBC in the ocean was linked to the emergence of aerobic metabolism, the acquisition of metabolic capabilities to exploit the newly created niches that followed the increase of oxygen, and the expansion of genes involved in the tolerance to salinity and oxidative stress.

The emergence of LBC (*Figures 3 and 4*), whose diversification occurred around the time of the NOE and the initial diversification of eukaryotic algae (*Parfrey et al., 2011*), was characterized by the enrichment of substantially different gene repertoires compared to EBC (*Figure 3*). For instance, the heterotrophic lineages Roseobacter, SAR116, and SAR92 show an enrichment of flagellar assembly and motility genes (*Figure 3*), including genes for flagellar biosynthesis, flagellin, and the flagellar basal-body assembly (*Supplementary file 4*). Motile marine heterotrophs like Roseobacter species have been associated with the marine phycosphere, a region surrounding individual phytoplankton cells releasing carbon-rich nutrients (*Mühlenbruch et al., 2018*; *Seymour et al., 2017*). Although the phycosphere can also be established between prokaryotic phytoplankton and heterotrophs (*Croft et al., 2005*; *Seymour et al., 2017*), given the late diversification of abundant marine prokaryotic phytoplankton (*Figures 2 and 4*), it is plausible that the emergence of these clades was closely related to the establishment of ecological proximity with large eukaryotic algae. The potential diversification of heterotrophic LBC due to their ecological interactions with eukaryotic algae is further supported by the enrichment of genes involved in vitamin B6 metabolism and folate biosynthesis, which are key nutrients involved in phytoplankton–bacteria associations (*Seymour et al., 2017*). LBC were also enriched in genes for the catabolism of taurine (e.g., taurine transport system permease in SAR11

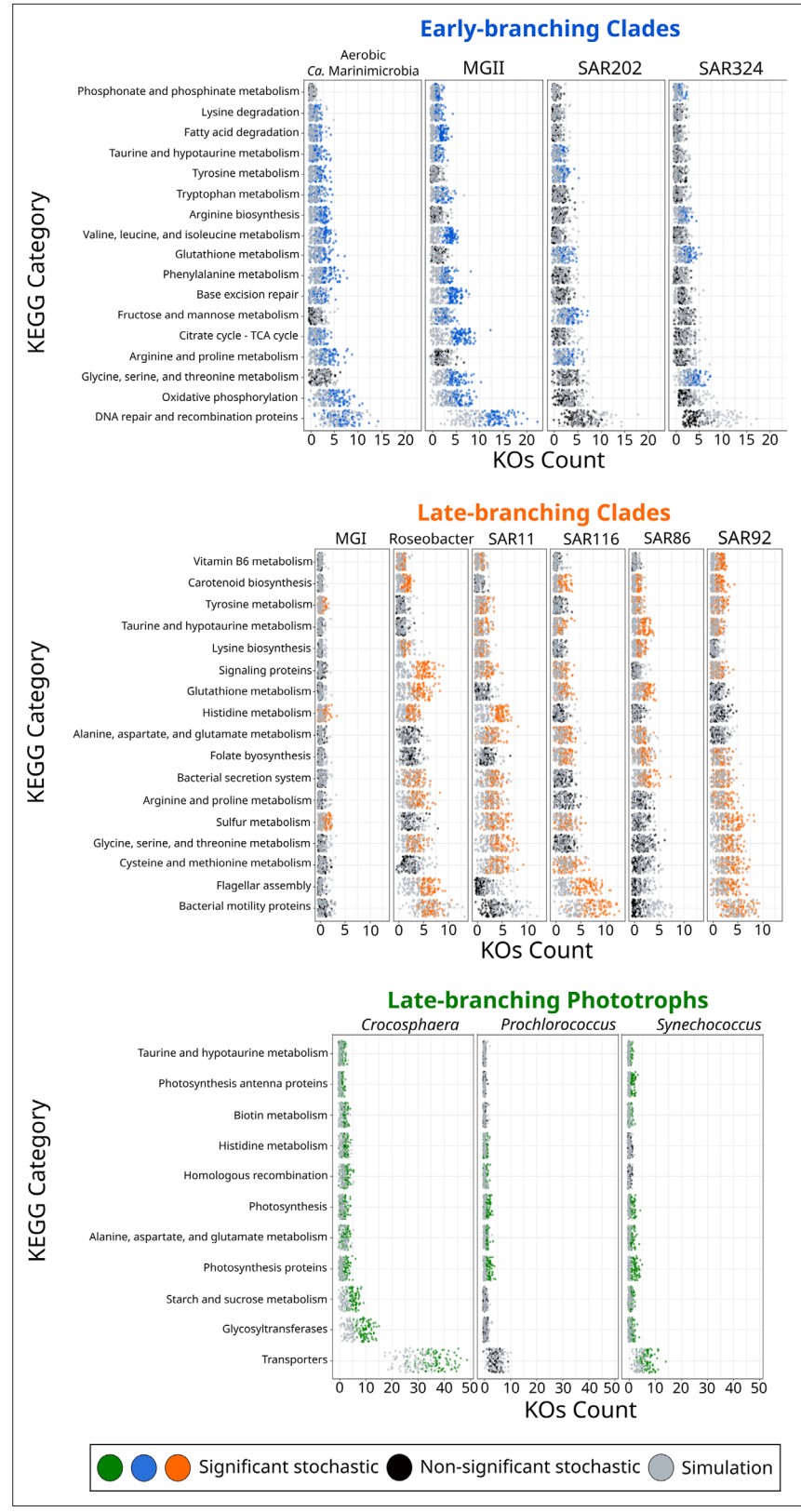

**Figure 3.** KEGG categories enriched at the crown node of each marine microbial clade. Clades were classified based on their diversification timing shown in *Figure 2*. Enriched categories were identified by statistically comparing a stochastic mapping distribution with an all-rates-different model vs a null distribution with a constant rate model without conditioning on the presence/absence data at the tips of the phylogeny. Each dot represents

*Figure 3 continued on next page*

*Figure 3 continued*

one replicate (see 'Materials and methods'). X-axis represents the number of KEGG Orthologous Groups (KOs) gained at each crown node for each KEGG category. Stochastic mapping and null distributions were sorted for visualization purposes. The complete list of enriched KEGGs is shown in ***Figure 3—figure supplement 1***.

The online version of this article includes the following figure supplement(s) for figure 3:

**Figure supplement 1.** Enriched KEGG categories at the crown node of each marine microbial clade.

and a taurine dioxygenase in SAR86 and SAR92), suggesting that LBC gained metabolic capabilities to utilize the taurine produced by other organisms as a substrate (***Clifford et al., 2019***), instead of producing it as an osmoprotectant. Furthermore, we identified the enrichment of genes involved in carotenoid biosynthesis, including spheroidene monooxygenase, carotenoid 1,2-hydratase, beta-carotene hydroxylase, and lycopene beta-cyclase (***Supplementary file 4***). The production of carotenoids is consistent with their use in proteorhodopsin, a light-driven proton pump that is a hallmark feature of most marine heterotrophic bacteria, in particular those that inhabit energy-depleted areas of the ocean today (***de la Torre et al., 2003***).

LBP that diversified around the time of the POE (***Figure 2***) showed a remarkable enrichment of transporters in *Crocosphaera* and *Synechococcus* (***Figure 3***). In particular, the diversification of *Crocosphaera* was characterized by the acquisition of transporters for inorganic nutrients like cobalt, nickel, iron, phosphonate, phosphate, ammonium, and magnesium, along with organic nutrients including amino acids and polysaccharides (***Supplementary file 4***). The acquisition of a wide diversity of transporters by the *Crocosphaera* is consistent with their boom-and-bust lifestyle seen in the oligotrophic open ocean today (***Hewson et al., 2009***; ***Wilson et al., 2017***), which requires a rapid and efficient use of scarce nutrients. We also identified genes involved in osmotic pressure tolerance, for example, a Ca-activated chloride channel homolog, a magnesium exporter, and a fluoride exporter (***Supplementary file 4***). In contrast, our results show that *Synechococcus* only acquired transporters for inorganic nutrients (e.g., iron and sulfate, ***Supplementary file 4***), whereas *Prochlorococcus* did not show the enrichment of transporters (***Supplementary file 4***). Similar to LBC, we identified the enrichment of taurine metabolism genes in *Crocosphaera* and *Synechococcus*, suggesting that its use as osmoprotectant and potential substrate is widespread among planktonic microorganisms (***Clifford et al., 2019***). *Prochlorococcus* exhibits enrichment in fewer categories than the rest of phototrophic clades diversifying during the same period, consistent with the streamlined nature of genomes from this lineage (***Partensky and Garczarek, 2010***). The genes acquired by *Prochlorococcus* are involved in photosynthesis, which supports previous findings that the diversification of this clade was accompanied by changes in the photosynthetic apparatus compared with *Synechococcus*, its sister group (***Biller et al., 2015***). Overall, the diversification of LBP was marked by the capacity to thrive in the oligotrophic ocean by exploiting organic and inorganic nutrients and by the modifying the photosynthetic apparatus as observed in *Crocosphaera* and *Synechococcus*, and *Prochlorococcus*, respectively.

## Conclusion

The contemporary ocean is dominated by abundant clades of bacteria and archaea that drive global biogeochemical cycles and play a central role in shaping the redox state of the planet. Despite their importance, the timing and geological landscape in which these clades colonized the ocean have remained unclear due to a combination of the inherent difficulties of studying biological events that occurred in deep time and the lack of a fossil record for microbial life. Yet establishing a timeline of these events is critical because the colonization of major marine lineages led to the establishment of the biogeochemical cycles that govern the environmental health of the planet today. In this study, we develop a novel phylogenomic method that allows us to infer a comprehensive timeline of the colonization of the ocean by abundant marine clades of both bacteria and archaea. Importantly, our study presents key foundational knowledge for understanding ongoing anthropogenic changes in the ocean. Climate change is predicted to lead to an expansion of both oxygen minimum zones, which our findings suggest are refugia that date back to the mid-Proterozoic ocean, and oligotrophic surface waters, which represent ecosystems that emerged relatively recently in the Phanerozoic (***Figure 4***). Thus, the impacts of current global change can manifest similarly in ecosystems that have emerged at dramatically different periods of Earth's history. Knowledge of how and under what geochemical conditions dominant microbial constituents first diversified provides context for understanding the

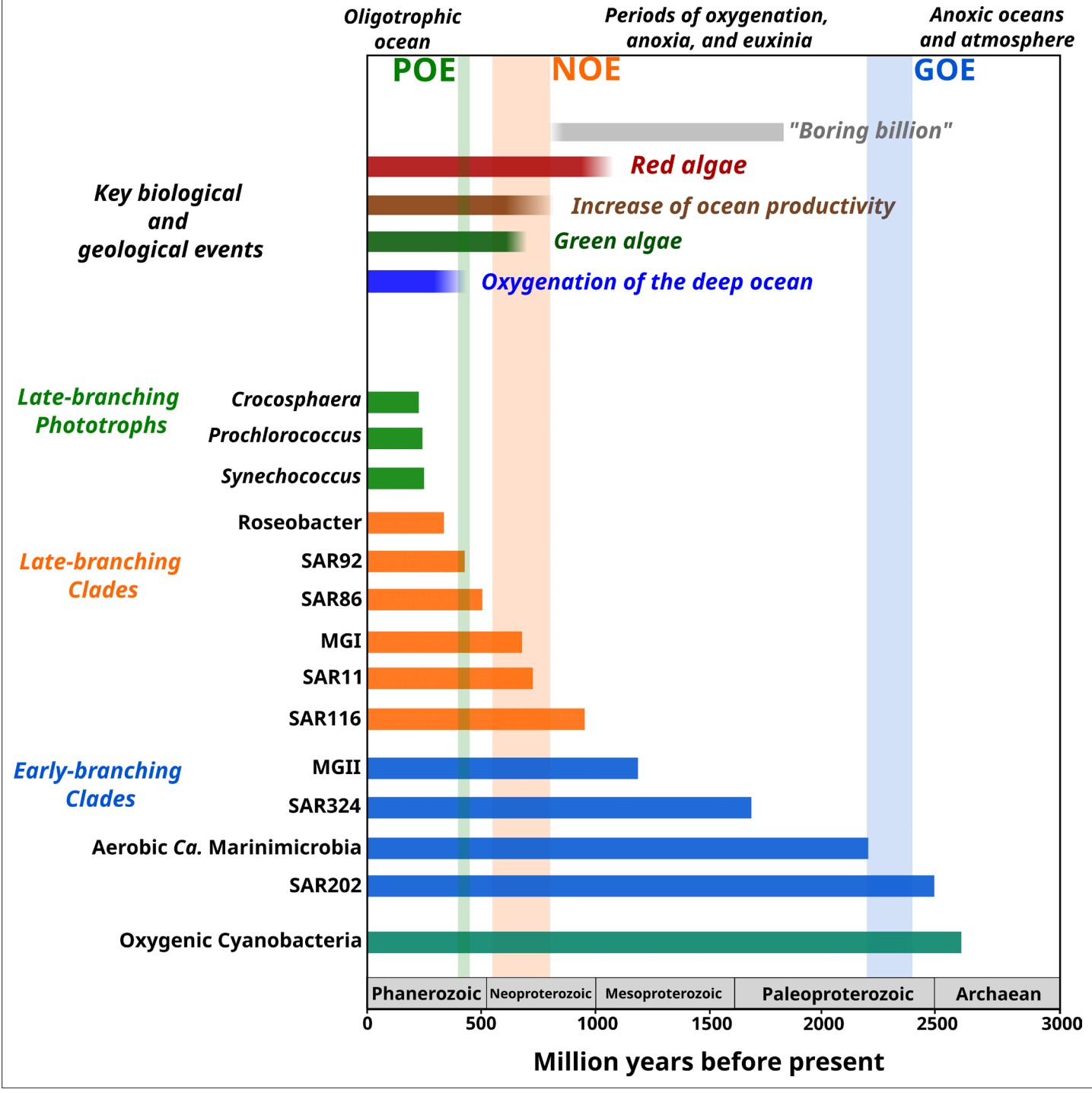

**Figure 4.** Link between the timing of the diversification of the main marine microbial clades and major geological and biological events. The timing of the geological and biological events potentially involved in the diversification of marine clades is based on previously published data: 'boring billion' (**Brasier and Lindsay, 1998**; **Hodgskiss et al., 2019**), red algae fossils (**Butterfield, 2000**), increased of ocean productivity (**Butterfield, 2000**; **Och and Shields-Zhou, 2012**), green algae fossils (**Butterfield et al., 2006**), and oxygenation of the deep ocean (**Lenton et al., 2016**). The length of each bar represents the estimated age for marine clades according to Bayesian estimates. The timing of the main oxygenation events is based on previous work (**Alcott et al., 2019**).

impact of climatic changes on the marine biome more broadly and will help clarify how continuing ecological shifts will impact marine biogeochemical cycles.

## Materials and methods

### Genome sampling and phylogenetic reconstruction

To obtain a comprehensive understanding of the diversification of the main marine planktonic clades, we built a multi-domain phylogenetic tree that included a broad diversity of bacterial and archaeal genomes. We compiled a balanced genome dataset from the Genome Taxonomy Database (GTDB, v95; *Chaumeil et al., 2019*), including marine representatives, by using a genome sampling strategy reported previously (*Martinez-Gutierrez and Aylward, 2021*). In addition, we improved the representation of marine clades by subsampling genomes from the GORG database (*Pachiadaki et al., 2019*), which includes a wide range of genomes derived from single-cell sequencing, and added several *Thermoarchaeota* genomes available on the JGI (*Nordberg et al., 2014*). We discarded genomes belonging to the DPANN superphylum due to the uncertainty of their placement within the archaea (*Martinez-Gutierrez and Aylward, 2021*). The list of genomes used is reported in *Supplementary file 1*.

We reconstructed a phylogenetic tree through the benchmarked MarkerFinder pipeline developed previously (*Martinez-Gutierrez and Aylward, 2021*), which resulted in an alignment of 27 ribosomal genes and 3 RNA polymerase genes (RNAP) (*Martinez-Gutierrez and Aylward, 2021*). The Marker-Finder pipeline consists of (1) the identification of ribosomal and RNAP genes using HMMER v3.2.1 with the reported model-specific cutoffs (*Eddy and Pearson, 2011*; *Sievers and Higgins, 2018*), (2) alignment with ClustalOmega (*Sievers and Higgins, 2018*), and (3) concatenation of individual alignments. The resulting concatenated alignment was trimmed using trimAl (*Capella-Gutiérrez et al., 2009*) with the option -gt 0.1. Phylogenetic tree inference was carried out with IQ-TREE v1.6.12 (*Nguyen et al., 2015*) with the options -wbt, -bb 1000 (*Minh et al., 2013*), -m LG +R10 (substitution model previously selected with the option -m MFP according to the Bayesian Information Criterion (BIC); *Kalyaanamoorthy et al., 2017*), and `--runs` 5 to select the tree with the highest likelihood. The tree with the highest likelihood was manually inspected to discard the presence of topological inconsistencies and artifacts on iTOL (*Letunic and Bork, 2019*; *Figure 1*). The raw phylogenetic tree is presented in *Supplementary file 2*. In a previous study, we assessed the effect of substitution model selection on the topology of a multi-domain phylogenetic tree (*Martinez-Gutierrez and Aylward, 2021*); however, we did not observe topological changes between the substitution models LG + C60 and LG + R10 (model selected according to the BIC criterion).

### Assessment of tree quality

Due to the key importance of tree quality for the tree-dependent analysis performed in our study, we assessed the congruence of our prokaryotic ToL through the Tree Certainty metric (TC) (*Martinez-Gutierrez and Aylward, 2021*; *Salichos et al., 2014*), which has recently been shown to be a more accurate estimate for phylogenetic congruence than the traditional bootstrap. Our estimate based on 1000 replicate trees (TC = 0.91) indicates high congruence in our phylogeny, indicating that the phylogenetic signal across our concatenated alignment of marker genes is consistent. We also evaluated whether the topology of our ToL agrees with a high-quality prokaryotic ToL reported previously (*Martinez-Gutierrez and Aylward, 2021*). In general, we observed consistency in the placement of all the phyla, as well as the bacterial superphyla (Terrabacteria and Gracilicutes) between both trees, except for the sisterhood of Actinobacteriota and Armatimonadota, which differs from the sisterhood of Actinobacteriota and Firmicutes in the reference tree (*Martinez-Gutierrez and Aylward, 2021*). We do not expect these discrepancies to substantially impact the results of our study because none of the marine clades are within this region of the tree.

### Estimating the age of the crown node of bacterial and archaeal marine clades

To investigate the timing of the diversification of the marine planktonic clades, the phylogenetic tree obtained above was used to perform a molecular dating analysis of the crown nodes leading to the diversification of the main marine microbial clades. We focused our analysis on clades of bacteria and

archaea that are overwhelmingly marine, such that the evolutionary history of that clade could be clearly traced back to an ancestral colonization of the ocean. Some clades, such as marine *Nitrospinae* and Actinobacteria were not considered because they included several non-marine members, and it was unclear whether these lineages colonized the ocean multiple times independently. Our analysis was performed through Phylobayes v4.1c (*Lartillot et al., 2009*) with the program pb on four independent chains. For each chain, the input consisted in the phylogenetic tree, the amino acid alignment, the calibrations, and an autocorrelated relaxed log normal model (-ln) (*Thorne et al., 1998*) with the molecular evolution model CAT-Poisson + G4. Convergence was tested every 5000 cycles using the program tracecomp with a burn-in of 250 cycles and sampling every 2 cycles. After 100,000 cycles, our chains reached convergence in 8 out of 12 parameters (*Supplementary file 3*). To assess the uncertainty derived from the parameters that did not reach convergence, we estimated the divergence ages with the program readdiv using the last 1000 cycles and testing every 10 cycles of the four chains of our Bayesian analysis (*Figure 2—figure supplement 1*). Although some Bayesian parameters did not reach convergence after 100,000 cycles , the estimated ages resulting from our four independent chains were similar when compared to each other (*Figure 2—figure supplement 1*). However, we observed an overall decrease in consistency between chains in the earliest clades (MGII, SAR324, Aerobic *Ca.* Marinimicrobia, and SAR202). This discrepancy is probably due to a decline in phylogenetic signal toward the root of the phylogenetic tree (*Philippe et al., 2011*).

## Selection of priors and assessment of priors' impact on posterior distribution

To determine the suitability and impact of our priors on the age estimates of the calibrated nodes, we ran an independent MCMC chain without the amino acid alignment using the option -root on Phylobayes. Our prior-only analysis yielded a posterior age falling within the maximum and minimum priors used for the crown group of archaea and bacteria. For the internally calibrated nodes, we observed posterior estimates consistent with the priors used for each case except for aerobic ammonia-oxidizing archaea (*Figure 2-figure supplement 2*). Overall, this result suggests that the calibrations used as priors were adequate for our analyses.

## Molecular dating analysis based on penalized likelihood and assessment of priors role on age estimates

We evaluated the reproducibility of our Bayesian divergence estimates by running an additional analysis based on penalized likelihood (PL) using TreePL (*Smith and O'Meara, 2012*) on 1000 replicate trees. Replicate trees were generated with the program bsBranchLenghts available on RAxML v8.2.12 (*Stamatakis, 2014*). For each of the 1000 replicate runs, we initially used the option 'prime' on TreePL to identify the optimization parameters and applied the parameter 'through' to continue iterations until parameter convergence. Moreover, we estimated the optional smoothing value for each replicate tree and ran cross-validation with the options 'cv' and 'randomcv.' The divergence times resulting from the 1000 bootstrap trees were used to assess the age variation for each marine microbial clade (*Figure 2—figure supplement 1*). Moreover, we used a PL approach to assess the role of calibrations on age estimates by using two different sets of priors. The first set consisted in using the priors shown in *Table 1* (Priors set 1) and the second set included the independent calibration of the bacterial and archaeal root and the calibration of the crown node of oxygenic Cyanobacteria (Priors set 2).

## Assessing the role of molecular dating strategy, molecular dating rate model, and calibrations on the diversification timing estimates of marine microbial clades

In order to evaluate the reproducibility of our Bayesian molecular dating analysis and assess the reliability of the calibration points used (*Figure 2*), we applied multiple additional molecular dating analyses. Firstly, using the same calibrations, we applied a second independent approach based on Penalized likelihood (PL) (*Smith and O'Meara, 2012*) (see previous section). We found consistency in the age estimates between PL and Bayesian except for the clades *Prochlorococcus* and *Synechococcus*, which showed a more recent diversification when using a Bayesian approach (*Figure 2—figure supplement 3*). Secondly, we evaluated the role of model selection on our Bayesian posterior estimates by running two additional Bayesian analyses under the relaxed molecular clock models CIR

(Autocorrelated CIR process; *Lepage et al., 2007*) and UGAM (Uncorrelated Gamma Multiples; *Drummond et al., 2006*) available on Phylobayes. Overall, our estimates once again revealed broad consistency across models, with the exception that SAR11 and SAR86 had notably earlier divergence times with the CIR and UGAM models relative to the log-normal model. Previous research has shown that autocorrelated models outperform uncorrelated models when tested in different datasets (*Lepage et al., 2007*), which would suggest that the autocorrelated log-normal and CIR models provide the most robust estimates in our analysis. Indeed, for SAR86 the UGAM model provided an unusually early diversification date that is an outlier compared to all other estimates (*Figure 2—figure supplement 3*).

Lastly, due to the potential limitations of using the oxygenation of the atmosphere (GOE) as a maximum prior for the strict aerobic metabolism of aerobic *Ca*. Marinimicrobia, ammonia-oxidizing archaea, and nitrate-oxidizing bacteria (*Table 1*), we performed an additional molecular dating analysis using a PL approach in which these priors were excluded (Priors set 2; *Figure 2—figure supplement 3*). Our analysis once again showed similar divergence times in all marine clades regardless of the priors used (*Figure 2—figure supplement 3*), indicating that the use of these calibrations did not strongly shape our results. Importantly, the overall consistency in our age estimates using different molecular dating approaches, models, and priors does not alter our main conclusions regarding the emergence of marine microbial clades and the geochemical context in which they first diversified.

## Comparing Bayesian diversification estimates with previous studies

Two estimated divergence times shown in our study disagree with previously published analyses. Firstly, a recent molecular dating estimate suggested that the transition of AOA-Archaea from terrestrial environments into marine reals occurred before the NOE (*Ren et al., 2019*) during a period known as the 'boring million' characterized by low productivity and minimum oxygen concentrations in the atmosphere (0.1% the present levels) (*Anbar and Knoll, 2002*; *Hodgskiss et al., 2019*; *Holland, 2006*; *Reinhard and Planavsky, 2022*). Our estimates point to a later diversification of this lineage during or after the NOE (678 My, 95% CI 668–688 My) (*Figure 2*), which is comparable with the age reported by another independent study (*Yang et al., 2021*). Secondly, another study reported the origin of the Picocyanobacterial clade *Prochlorococcus* to be 800 My, before the Snowball Earth period registered during the Cryogen (*Zhang et al., 2021*). However, our results agree with another independent study that points to a relatively late evolution of *Prochlorococcus* (*Sánchez-Baracaldo, 2015*).

## Orthologous groups detection, stochastic mapping, and functional annotation

To investigate the genomic novelties associated with the diversification of the marine microbial lineages considered in our study, we identified enriched KEGG categories in the crown node of each clade. First, we predicted protein orthologous groups with ProteinOrtho v6 (*Lechner et al., 2011*) using the option 'lastp' and the protein files downloaded from the GTDB, GORG, and JGI databases. Furthermore, we performed a functional annotation using the KEGG database (*Kanehisa, 2019*; *Kanehisa et al., 2021*; *Kanehisa and Goto, 2000*) through HMMER3 with an e-value of $10^{-5}$ on all proteins. Proteins with multiple annotations were filtered to keep the best-scored annotation, and we predicted the function of each protein orthologous group by using the Majority Rule Principle. The presence/absence matrix resulting from the identification of orthologous groups was used together with our phylogenetic tree to perform 100 replicate stochastic mapping analyses on each orthologous group with the make.simmap function implemented on Phytools (*Bollback, 2006*; *Revell, 2012*) with the model 'all-rates-different' (ARD). To evaluate evidence of enrichment of KEGG categories, we simulated a null distribution for each protein cluster under the transition matrix estimated from our stochastic mapping analysis using the function sim.history, but without conditioning on the presence/absence data at the tips (i.e., simulating a constant rate null distribution of transitions across the tree). Since some of the protein clusters show a low exchange rate (identified because one of the rows in the Q-matrix was equal to zero), we manually changed the exchange rate from 0 to 0.00001. For each distribution, we estimated the number of genes gained for each KEGG category at the crown node of the marine clades. Clusters without a known annotation on the KEGG database were discarded. The resulting KEGG category distributions for our stochastic mapping and null analyses were statistically

compared using a one-tailed Wilcoxon test ($\alpha = 0.01$, N = 100 for each distribution). KEGG categories showing statistically more gains in our stochastic mapping distribution were considered enriched (*Figure 3—figure supplement 1*).

## Acknowledgements

We acknowledge the use of the Virginia Tech Advanced Research Computing Center for bioinformatic analyses performed in this study. This investigation was supported by grants from the National Science Foundation (IIBR-2141862) and a Simons Foundation Early Career Award in Marine Microbial Ecology and Evolution to FOA. We kindly thank members of the Aylward Lab for their insightful comments on an earlier version of this manuscript.

## Additional information

### Funding

| Funder | Grant reference number | Author |
|---|---|---|
| National Science Foundation | IIBR-2141862 | Frank O Aylward |
| Simons Foundation | Early Career Award in Marine Microbial Ecology and Evolution | Frank O Aylward |

The funders had no role in study design, data collection and interpretation, or the decision to submit the work for publication.

### Author contributions

Carolina A Martinez-Gutierrez, Conceptualization, Resources, Data curation, Software, Formal analysis, Validation, Investigation, Visualization, Methodology, Writing – original draft, Writing – review and editing; Josef C Uyeda, Conceptualization, Resources, Formal analysis, Investigation, Methodology, Writing – original draft; Frank O Aylward, Conceptualization, Resources, Data curation, Software, Formal analysis, Supervision, Funding acquisition, Validation, Investigation, Visualization, Methodology, Writing – original draft, Project administration, Writing – review and editing

### Author ORCIDs

Carolina A Martinez-Gutierrez ● https://orcid.org/0000-0002-8365-6050
Frank O Aylward ● https://orcid.org/0000-0002-1279-4050

Reviewer #1 (Public Review): https://doi.org/10.7554/eLife.88268.3.sa1
Reviewer #2 (Public Review): https://doi.org/10.7554/eLife.88268.3.sa2
Author Response https://doi.org/10.7554/eLife.88268.3.sa3

## Additional files

### Supplementary files

• Supplementary file 1. Genomes dataset used for the molecular dating of the main marine microbial clades.

• Supplementary file 2. Raw maximum likelihood phylogenetic tree used for molecular dating and stochastic mapping analyses.

• Supplementary file 3. Assessment of parameters convergence of four independent chains used for Bayesian molecular dating analyses. Relative difference < 0.3 is shown in bold letters and denotes parameters that reached convergence after 100,000 cycles using a burn-in of 250 and sampling every two cycles.

• Supplementary file 4. KOs gained at the crown node of each marine microbial clade. A KO was considered as gained when found in 51 out of 100 stochastic mapping replicates.

• Supplementary file 5. Age estimates of marine microbial clades resulting from different Bayesian molecular dating models (log-normal, CIR, and UGAM) and calibrations (TreePL priors set 1 and 2). Bayesian estimates represent the average of the last 1000 cycles sampled every 10 cycles of each of the four chains. TreePL analyses show 1000 age replicates using the priors shown in *Table 1* (Priors set 1), and the independent root of Bacteria and Archaea and the minimum age of Cyanobacteria as priors (Priors set 2).

• MDAR checklist

### Data availability

The main code used in our study is deposited on GitHub: https://github.com/carolinaamg/enriched_OG (copy archived at *Carolinaamg, 2023*).

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
