## [Editor Report · eLife assessment]

This **important** paper addresses the challenging problem of dating the origin of several groups of marine microorganisms. The analyses are **solid**, with various test of clock models and time calibrations used; however, given the uncertainty of many of the dates used to anchor ancient geological events, further studies are needed to support or refute the hypotheses put forth in this paper. Despite some methodological concerns, this work is a commendable attempt at an extremely difficult problem and will be of broad interest to microbiologists, geologists, and evolutionary biologists.

---

## [Referee Report · Reviewer #1 (Public Review)]

Martinez-Gutierrez and colleagues presented a timeline of important bacteria and archaea groups in the ocean and based on this they correlated the emergence of these microbes with GOE and NOE, the two most important geological events leading to the oxygen accumulation of the Earth.

The following suggestion is very important and requires additional clock analysis.

"Three calibrations at Aerobic Nitrososphaerales, Aerobic Marinimicrobia, and Nitrite oxidizing bacteria have the same problem - they are all assumed to have evolved after the GOE where the Earth started to accumulate oxygen in the atmosphere, so they were all capped at 2320 Ma. This is an important mistake and will significantly affect the age estimates because maximum constraint was used (maximum constraint has a much greater effect on age estimates and minimum constraint), and this was used in three nodes involving both Bacteria and Archaea. The main problem is that the authors ignored the numerous evidence showing that oxygen can be produced far before GOE by degradation of abiotically-produced abundant H2O2 by catalases equipped in many anaerobes, also produced by oxygenic cyanobacteria evolved at least 500 Ma earlier than the onset of GOE (2500 Ma), and even accumulated locally (oxygen oasis). It is well possible that aerobic microbes could have evolved in the Archaean."

---

## [Referee Report · Reviewer #2 (Public Review)]

In this paper, Martinez-Gutierrez and colleagues present a dated, multidomain ( = Archaea+Bacteria) phylogenetic tree, and use their analyses to directly compare the ages of various marine prokaryotic groups. They also perform ancestral gene content reconstruction using stochastic mapping to determine when particular types of genes evolved in marine groups.

Overall, there are not very many papers that attempt to infer a dated tree of all prokaryotes, and this is a distinctive and up-to-date new contribution to that oeuvre. There are several particularly novel and interesting aspects - for example, using the GOE as a (soft) maximum age for certain groups of strictly aerobic Bacteria, and using gene content enrichment to try to understand why and how particular marine groups radiated.

One overall feature of the results is that marine groups tend to be quite young, and there don't seem to be any modern marine groups that were in the ocean prior to the GOE. This seems an interesting strand to pursue in future work. Presumably, the earliest branches of the bacterial tree were marine, so what happened in the intervening period? The authors' character mapping approach could also be used to infer the habitat of the Gracilicutes and Terrabacteria ancestors, and it might be interesting to revisit the question of the ancestral ecological differences between these groups, if any can be clearly distinguished.

Finally, some comments in which I disagree with a couple of the authors' methodological decisions. I don't think these disagreements are likely to have a major impact on the findings, but I feel it is worth mentioning them in any case, to stimulate future discussion and work. I very much appreciate that finding time calibrations for microbes is challenging, but I nonetheless have a couple of comments or concerns about the calibrations used here.

1. It is not clear that the earliest evidence for biogenic methane provides a minimum age for both Bacteria and Archaea. For Archaea, potentially --- if the methane is indeed biogenic, and if the last archaeal common ancestor was a methanogen. For Bacteria (and extant life as a whole), the link is harder to draw. The authors pointed out that there is other evidence from around this time for life, for example from the Strelley Pool at ~3.3Ga. This is a reasonable argument for a minimum on LUCA, but then the optimal approach would be to calibrate the root node with this minimum, rather than the two descendant clades.

2. I am also unclear about the rationale for setting the minimum age of the photosynthetic Cyanobacteria crown to the time of the GOE. Presumably, oxygen-generating photosynthesis evolved on the stem of (photosynthetic) Cyanobacteria - since the crown seems to have had it ancestrally - and it therefore seems possible that the GOE might have been initiated by these stem Cyanobacteria, with the crown radiating later. In their response to my comment, the authors confirm that they are calibrating the crown Cyanobacteria using the GOE as a minimum. I don't agree with the logic here: it seems a formal possibility that crown Cyanobacteriia are younger than the GOE. The authors argued that, although oxygenic photosynthesis likely evolved on the stem, due to extinction (or non-sampling) of intervening lineages there are no nodes on the tree that directly sample that event. I agree, but I would then suggest placing the minimum on the older, not the younger, end of the stem.

---

## [Author Response]

The following is the authors’ response to the original reviews.

Thank you for your time and effort in handling and reviewing our manuscript. We have responded to all comments below.

**Reviewer #1 (Public Review):**
Martinez-Gutierrez and colleagues presented a timeline of important bacteria and archaea groups in the ocean and based on this they correlated the emergence of these microbes with GOE and NOE, the two most important geological events leading to the oxygen accumulation of the Earth. The whole study builds on molecular clock analysis, but unfortunately, the clock analysis contains important errors in the calibration information the study used, and is also oversimplified, leaving many alternative parameters that are known to affect the posterior age estimates untested. Therefore, the main conclusion that the oxygen availability and redox state of the ocean is the main driver of marine microbial diversification is not convincing.

We do not conclude that “oxygen availability and redox state of the ocean is the main driver of marine microbial diversification”. Our conclusion is much more nuanced. We merely discuss our findings in light of the major oxygenation events and oxygen availability (among other things) given the important role this molecule has played in shaping the redox state of the ocean.

Regarding the methodological concerns, to address them we have provided additional analyses to account for different clock models and calibration points.

Basically, what the molecular clock does is to propagate the temporal information of the nodes with time calibrations to the remaining nodes of the phylogenetic tree. So, the first and the most important step is to set the time constraints appropriately. But four of the six calibrations used in this study are debatable and even wrong.(1) The record for biogenic methane at 3460 Ma is not reliable. The authors cited Ueno et al. 2006, but that study was based on carbon isotope, which is insufficient to demonstrate biogenicity, as mentioned by Alleon and Summons 2019.

Thank you for pointing out the limitations of using the geochemical evidence of methane as calibrations. Indeed, several commentaries have suggested that the biotic and abiotic origin of the methane reported by Ueno et al. are equally plausible (Alleon and Summons, 2019; Lollar and McCollom, 2006), however; we used that calibration as a minimum for the presence of life on Earth, not methanogenesis. Despite the controversy regarding the origin of methane, there are other lines of evidence suggesting the presence of life around ~3.4 Ga. For example stromatolites from the Dresser Formation, Pilbara, Western Australia (Djokic et al., 2017; Walter et al., 1980; Buick and Dunlop, 1990), and more recently (Hickman-Lewis et al., 2022). To avoid confusion, we have added a more extended explanation for the use of that calibration and additional evidence of life around that time in Table 1 and lines 100-104.

(2) Three calibrations at Aerobic Nitrososphaerales, Aerobic Marinimicrobia, and Nitrite oxidizing bacteria have the same problem - they are all assumed to have evolved after the GOE where the Earth started to accumulate oxygen in the atmosphere, so they were all capped at 2320 Ma. This is an important mistake and will significantly affect the age estimates because maximum constraint was used (maximum constraint has a much greater effect on age estimates and minimum constraint), and this was used in three nodes involving both Bacteria and Archaea. The main problem is that the authors ignored the numerous evidence showing that oxygen can be produced far before GOE by degradation of abiotically-produced abundant H2O2 by catalases equipped in many anaerobes, also produced by oxygenic cyanobacteria evolved at least 500 Ma earlier than the onset of GOE (2500 Ma), and even accumulated locally (oxygen oasis). It is well possible that aerobic microbes could have evolved in the Archaean.

We appreciate the suggestion of assessing the validity of the calibrations used in our analyses. We initially evaluated the informative power of the priors used for the Bayesian molecular dating (Supplemental File 5), and found that the only calibration that lacked enough information for the purposes of our study was Ammonia Oxidizing Archaea (AOA). In contrast to previous evidence (Ren et al., 2019; Yang et al., 2021), we associate this finding to the potential earlier diversification of AOA. Due to the limitations of several of the calibrations used, we performed an additional molecular dating analysis on 1000 replicate trees using a Penalized Likelihood strategy. This analysis consisted in excluding the calibrations that assumed the presence of oxygen as a maximum constraint. Our analysis shows similar age estimates of the marine microbial clades regardless of the exclusion of these calibrations (Supplemental File 8; TreePL Priors set 2). Our findings thus suggest that the age estimates reported in our study are consistent regardless of whether or not the presence of oxygen is used to calibrate several nodes in the tree. We describe the results of this analysis in lines 490-499 and include estimates in Supplemental File 8. Our results are therefore robust regardless of the use of these somewhat controversial calibrations.

Once the phylogenetic tree is appropriately calibrated with fossils and other time constraints, the next important step is to test different clock models and other factors that are known to significantly affect the posterior age estimates. For example, different genes vary in evolutionary history and evolutionary rate, which often give very different age estimates. So it is very important to demonstrate that these concerns are taken into account. These are done in many careful molecular dating studies but missing in this study.

We agree that the selection of marker genes will have a profound impact on the final age estimates. First, it is important to understand that very few genes present in modern Bacteria and Archaea can be traced back to the Last Universal Common Ancestor, so there are very few genes to use for this purpose. Studies that focus on particular groups of Bacteria and Archaea may have larger selections of genes to choose from, but for our purposes there are only about ~40 different genes - mostly encoding for ribosomal proteins, RNA polymerase subunits, and tRNA synthetases - that can be use for this purpose (Creevey et al., 2011; Wu and Scott, 2012). In a previous study we have extensively benchmarked methods for the reconstruction of high-resolution phylogenetic trees of Bacteria and Archaea using these genes (Martinez-Gutierrez and Aylward, 2021). Our analyses demonstrated that some of these genes (mainly tRNA synthetases) have undergone ancient lateral gene transfer events and are not suitable for deep phylogenetics or molecular dating. In this previous study we also evaluated different sets of marker genes to examine which provide the most robust phylogenetic inference. We arrived at a set of ribosomal proteins and RNA polymerase subunits that performs best for phylogenetic reconstruction, and we have used that in the current study.

Furthermore, we tested the role of molecular dating model selection on the final Bayesian estimates by running four independent chains under the models UGAM and CIR, respectively. Overall, the results did not vary substantially compared with the ages obtained using the log-normal model reported on our manuscript (Supplemental File 8). The additional results are described in lines 478-488 and shown in Supplemental File 8. The clades that showed more variation when using different Bayesian models were SAR86, SAR11, and Crown Cyanobacteria (Supplemental File 8). Despite observing some differences in the age estimates when using different molecular models, the conclusion that the different marine microbial clades presented in our study diversified during distinct periods of Earth’s history remains. Moreover, the main goal of our study is to provide a relative timeline of the diversification of abundant marine microbial clades without focusing on absolute dates.

**Reviewer #2 (Public Review):**
In this paper, Martinez-Gutierrez and colleagues present a dated, multidomain ( = Archaea+Bacteria) phylogenetic tree, and use their analyses to directly compare the ages of various marine prokaryotic groups. They also perform ancestral gene content reconstruction using stochastic mapping to determine when particular types of genes evolved in marine groups.Overall, there are not very many papers that attempt to infer a dated tree of all prokaryotes, and this is a distinctive and up-to-date new contribution to that oeuvre. There are several particularly novel and interesting aspects - for example, using the GOE as a (soft) maximum age for certain groups of strictly aerobic Bacteria, and using gene content enrichment to try to understand why and how particular marine groups radiated.

Thank you for your thorough evaluation and comments on our manuscript.

CommentsOne overall feature of the results is that marine groups tend to be quite young, and there don't seem to be any modern marine groups that were in the ocean prior to the GOE. It might be interesting to study the evolution of the marine phenotype itself over time; presumably some of the earlier branches were marine? What was the criterion for picking out the major groups being discussed in the paper? My (limited) understanding is that the earliest prokaryotes, potentially including LUCA, LBCA and LACA, was likely marine, in the sense that there would not yet have been any land above sea level at such times. This might merit discussion in the paper. Might there have been earlier exclusively marine groups that went extinct at some point?

Thank you for pointing this out - this is a very interesting idea.

Firstly, the major marine lineages that we study here have largely already been defined in previous studies and are known to account for a large fraction of the total diversity and biomass of prokaryotes in the ocean. For example, Giovannoni and Stingl described most of these groups previously when discussing cosmopolitan and abundant marine lineages (Giovannoni and Stingl, 2005). The main criteria to select the marine clades studied here are (1) these groups have large impacts in the marine biogeochemical cycles and represent a large fraction of the microbial biomass in the open ocean, (2) they have an appropriate representation on genomic databases such that they can be confidently included in a phylogenetic tree, (3) the clades included can be confidently classified as being marine, in the sense that consequently the last common ancestor had a marine origin. This is explained in lines 83-86. We were primarily interested in lineages that encompassed a broad phylogenetic breadth, and we therefore did not include many groups that can be found in the ocean but are also readily isolated from a range of other environments (i.e., Pseudomonas spp., some Actinomycetes, etc.).

We agree that some of the earlier microbial branches in the Tree of Life were likely marine. The study of the marine origin of LUCA, LBCA, LACA, although interesting, is out of the scope of our study, and our results cannot offer any direct evidence of their habitat. We have therefore sought to focus on the origins of extant marine lineages.

What do the stochastic mapping analyses indicate about the respective ancestors of Gracilicutes and Terrabacteria? At least in the latter case, the original hypothesis for the group was that they possessed adaptations to life on land - which seems connected/relevant to the idea of radiating into the sea discussed here - so it might be interesting to discuss what your analyses say about that idea.

Thank you for your recommendation to perform additional analysis regarding the characterization of the ancestor of the superphyla Gracilicutes and Terrabacteria. We agree that this analysis would be very interesting, but we wish to focus the manuscript primarily on the marine clades in question, and other supergroups are listed in Figure 2 mainly for context. However, we did check the results of the stochastic mapping analysis and we now report the list of genes predicted to be gained and lost at the ancestor of the Gracilicutes and Terrabacteria clades, however; it is out of the scope of this study.

I very much appreciate that finding time calibrations for microbes is challenging, but I nonetheless have a couple of comments or concerns about the calibrations used here:The minimum age for LBCA and LACA (Nodes 1 and 2 in Fig. 1) was calibrated with the earliest evidence of biogenic methane ~3.4Ga. In the case of LACA, I suppose this reflects the view that LACA was a methanogen, which is certainly plausible although perhaps not established with certainty. However, I'm less clear about the logic of calibrating the minimum age of Bacteria using this evidence, as I am not aware that there is much evidence that LBCA was a methanogen. Perhaps the line of reasoning here could be stated more explicitly. An alternative, slightly younger minimum age for Bacteria could perhaps be obtained from isotope data ~3.2Ga consistent with Cyanobacteria (e.g., see https://pubmed.ncbi.nlm.nih.gov/30127539/).

Thank you for pointing this out. We used the presence of methane as a minimum for life on Earth, not as a minimum for methanogenesis. Despite using this calibration as a minimum for the root of Bacteria and not having methanogenic representatives within this domain, there are independent lines of evidence that point to the presence of microbial life around the same time (~3.5 Ga, for example stromatolites from the Dresser Formation, Pilbara, Western Australia (~3.5 Ga) (Djokic et al., 2017; Walter et al., 1980; Buick and Dunlop, 1990)), and more recently (Hickman-Lewis et al., 2022). We added a rationale for the use of the evidence of methane as a minimum age for life on Earth to the manuscript (Table 1 and 100104).

I am also unclear about the rationale for setting the minimum age of the photosynthetic Cyanobacteria crown to the time of the GOE. Presumably, oxygen-generating photosynthesis evolved on the stem of (photosynthetic) Cyanobacteria, and it therefore seems possible that the GOE might have been initiated by these stem Cyanobacteria, with the crown radiating later? My confusion here might be a comprehension error on my part - it is possible that in fact one node "deeper" than the crown was being calibrated here, which was not entirely clear to me from Figure 1. Perhaps mapping the node numbers directly to the node, rather than a connected branch, would help? (I am assuming, based on nodes 1 and 2, that the labels are being placed on the branch directly antecedent to the node of interest)?

Thank you so much for your suggestion. As pointed out, the calibrations used were applied at the crown node of existing Cyanobacterial clades, not at the stem of photosynthetic Cyanobacteria. We agree that photosynthesis and therefore the production of molecular oxygen may have been present in more ancient Cyanobacterial clades, however; these groups have not been discovered yet or went extinct. We have improved Fig. 1 to avoid confusion and now it is part of the updated version of our manuscript.

Alleon J, Summons RE. 2019. Organic geochemical approaches to understanding early life. Free Radic Biol Med 140:103–112.

Buick R, Dunlop JSR. 1990. Evaporitic sediments of Early Archaean age from the Warrawoona Group, North Pole, Western Australia. Sedimentology 37: 247-277.

Creevey CJ, Doerks T, Fitzpatrick DA, Raes J, Bork P. 2011. Universally distributed single-copy genes indicate a constant rate of horizontal transfer. PLoS One 6:e22099.

Djokic T, Van Kranendonk MJ, Campbell KA, Walter MR, Ward CR. 2017. Earliest signs of life on land preserved in ca. 3.5 Ga hot spring deposits. Nat Commun 8:15263.

Giovannoni SJ, Stingl U. 2005. Molecular diversity and ecology of microbial plankton. Nature 437: 343-348.Hickman-Lewis K, Cavalazzi B, Giannoukos K, D'Amico L, Vrbaski S, Saccomano G, et al. 2023. Advanced two-and three-dimensional insights into Earth's oldest stromatolites (ca. 3.5 Ga): Prospects for the search for life on Mars. Geology 51: 33-38.

Lollar BS, McCollom TM. 2006. Geochemistry: biosignatures and abiotic constraints on early life. Nature.Martinez-Gutierrez CA, Aylward FO. 2021. Phylogenetic Signal, Congruence, and Uncertainty across Bacteria and Archaea. Mol Biol Evol 38:5514–5527.

Ren M, Feng X, Huang Y, Wang H, Hu Z, Clingenpeel S, Swan BK, Fonseca MM, Posada D, Stepanauskas R, Hollibaugh JT, Foster PG, Woyke T, Luo H. 2019. Phylogenomics suggests oxygen availability as a driving force in Thaumarchaeota evolution. ISME J 13:2150–2161.

Walter M R, R Buick, JSR Dunlop. 1980. Stromatolites 3,400–3,500 Myr old from the North pole area, Western Australia. Nature 284: 443-445.

Wu M, Scott AJ. 2012. Phylogenomic analysis of bacterial and archaeal sequences with AMPHORA2. Bioinformatics 28:1033–1034.

Yang Y, Zhang C, Lenton TM, Yan X, Zhu M, Zhou M, Tao J, Phelps TJ, Cao Z. 2021. The Evolution Pathway of Ammonia-Oxidizing Archaea Shaped by Major Geological Events. Mol Biol Evol 38:3637–3648.